# Changes in Land Use and Cover and Their Environmental Impacts in the Cerrado of Mato Grosso Do Sul, Brazil

Melina Fushimi [1], Gabriela Narcizo de Lima [2,3,*] and Viviane Capoane [4]

1. Department of Geography and Environmental Planning, Institute of Geosciences and Exact Sciences, São Paulo State University, Avenida 24 A, 1515, Rio Claro 13506-900, São Paulo, Brazil; melina.fushimi@unesp.br
2. Department of Geography, Porto University, Via Panorâmica, Campo Alegre, 4150-564 Porto, Portugal
3. Centre of Studies in Geography and Spatial Planning, Porto University, 4150-564 Porto, Portugal
4. Department of Geography, University of Mato Grosso do Sul State, Avenida Dom Antônio Barbosa, Campo Grande 79115-898, Mato Grosso do Sul, Brazil; viviane.capoane@uems.br
* Correspondence: gabrielalima@letras.up.pt

**Abstract:** In Brazilian regional landscapes, the Cerrado has one of the richest flora among the savannas in the world, with a high level of endemism; however, many plant species are threatened with extinction as a consequence of spatio-temporal changes in land use and cover. This study aimed to analyze changes in land use and cover in the upper course of the Ceroula stream basin, located in the Cerrado of Mato Grosso do Sul, Brazil, based on maps of land use and cover in 1985 and 2022, the normalized difference vegetation index (NDVI), precipitation data, and fieldwork. The results indicated that in 1985, forest vegetation was replaced by pasture, and in 2022, in addition to pasture, there was the introduction of soybean monoculture with corn in the off-season, influenced by the international commodities market. These land use and cover alterations, without adequate management and in the absence of conservation practices, led to environmental impacts, such as accelerated linear erosive processes (rill, ravine, and gully). The results may help provide important insights into the dynamics of land use and cover, the consequences of the lack of conservation practices, and the environmental impacts in the Cerrado of Mato Grosso do Sul, contributing to better understanding of the environmental challenges faced in the region and the need to provide subsidies for the development of sustainable management strategies.

**Keywords:** Cerrado; land use and cover; environmental impacts

## 1. Introduction

In Brazilian regional landscapes, the Cerrado presents one of the richest flora among the savannas in the world, with a high level of endemism (44% of the flora is endemic); however, many plant species are threatened with extinction. The main causes are the degradation of different types of vegetation, biological invasion caused by the planting of grasses of African origin (Brachiaria grass) that cover an area of 500,000 km$^2$ (equivalent to the area of Spain), monocultures (mainly soybeans) covering 184,662.98 km$^2$, deforestation, with rates varying between 22,000 and 30,000 km$^2$ per year, and accelerated erosion [1–3]. The main challenges faced by humanity include the intensive use of tropical savannas, especially for the production of food and energy, and resolution of these issues has a strong relationship with land use and cover [4].

The Brazilian Institute of Geography and Statistics [5] defines "land use" as human activities that make use of land resources associated with the socioeconomic function of the surface, and "land cover" as the elements of nature (e.g., natural and planted vegetation, water, and soil) and the constructions created by social practices that cover the surface. Methodologically, the survey of land use and cover indicates the geographical distribution of the typology of use, identified through homogeneous patterns of land cover, involving

office research and fieldwork to interpret, analyze, and record the types of land use and cover, with the preparation of maps [5].

According to the MapBiomas Project collection 7.1 [3], between 1985 and 2021, the Cerrado presented reductions in the savanna formation of 24.9% (from 805,385.94 km$^2$ to 605,135.70 km$^2$), in forest of 15.3% (from 331,223.04 km$^2$ to 280,526.98 km$^2$), and in non-forest natural formation of 13.9% (from 196,799.86 km$^2$ to 169,509.12 km$^2$). The agriculture and pasture areas increased, from 619,956.40 km$^2$ to 899,113.12 km$^2$ for agriculture and 382,219.98 km$^2$ to 470,118.64 km$^2$ for pasture. Soybeans (*Glycine max*) represented the biggest increase in the period, 1401.5% (from 12,298.67 km$^2$ in 1985 to 184,662.98 km$^2$ in 2021), followed by sugarcane, 978.9% (from 2582.55 km$^2$ in 1985 to 27,863.24 km$^2$ in 2021). Almost half of the native vegetation in the Cerrado has been transformed into soybeans or pasture.

In the Cerrado of Mato Grosso do Sul, the scenario is similar considering the same period (1985–2021), with a decrease in forest area (from 53,641.36 km$^2$ to 36,689.70 km$^2$) and an increase in agriculture (from 156,110.60 km$^2$ to 212,380.26 km$^2$) [3].

Considering that the land use and cover are dynamic, changing over historical time under the strong influence of the market (commodities), and that the absence or low adoption of management and sustainable conservation practices trigger processes of environmental degradation and as a consequence impact nature and society, the current study aimed to analyze changes in land use and cover in the upper course of the Ceroula stream basin, located in the Cerrado of Mato Grosso do Sul, Brazil.

The study aims to understand how changes in land use and cover occurred over historical time, especially considering the strong influence of the commodities market in this region. Furthermore, the research seeks to investigate how land use and cover, without sustainable management, can trigger environmental degradation processes, resulting in environmental impacts.

## 2. Location and Characterization of the Study Area

The upper course of the Ceroula stream basin has an area of 100.83 km$^2$, is located in the far north of the municipality of Campo Grande, capital of the state of Mato Grosso do Sul, Brazil, and is part of the Cerrado biome (Figure 1). The Ceroula stream and its tributaries make up the ecological corridor of the Intermunicipal Consortium for the Integrated Development of the Miranda and Apa river basins, which are important in the formation of the Pantanal floodplain [6] and are integrally inserted in the Environmental Protection Area of the Ceroula Stream (APA of Ceroula), established by Decree No. 8264 of 27 July 2001 [7].

The rocky substrate is composed of basalts from the Serra Geral Formation and sandstones from the Caiuá Formation [8]. The regional relief is characterized by plateaus, specifically the Dissected Plateau of the Western Edge of the Paraná Basin, the Sul-Matogrossense Plateau, and the Campo Grande Plateau [8]. In the study area, Fushimi and Capoane (2023) [9] identified three main relief compartments: 1. undulating tops of hills with slopes below 8%; 2. concave, convex, and rectilinear slopes, with slopes between 8% and 45% (scarp); 3. and floodplain and alveoli with slopes of up to 8%. The predominant soils are dystrophic red latosols and eutrophic litholic neosols [8].

The climate of the Brazilian Cerrado is humid tropical, featuring a rainy season in summer and a dry season in winter. This pattern is influenced by three main air masses: the Atlantic Polar Mass coming from the south; the Continental Equatorial Mass originating from the north; and the Tropical Continental Mass, which forms in the Chaco Low region, between the Amazon Basin and northern Argentina. Precipitation in the study area is predominantly affected by continental-origin low-pressure systems, especially by the Tropical Continental Mass, as highlighted by all atmospheric systems operating in the central south of Brazil.

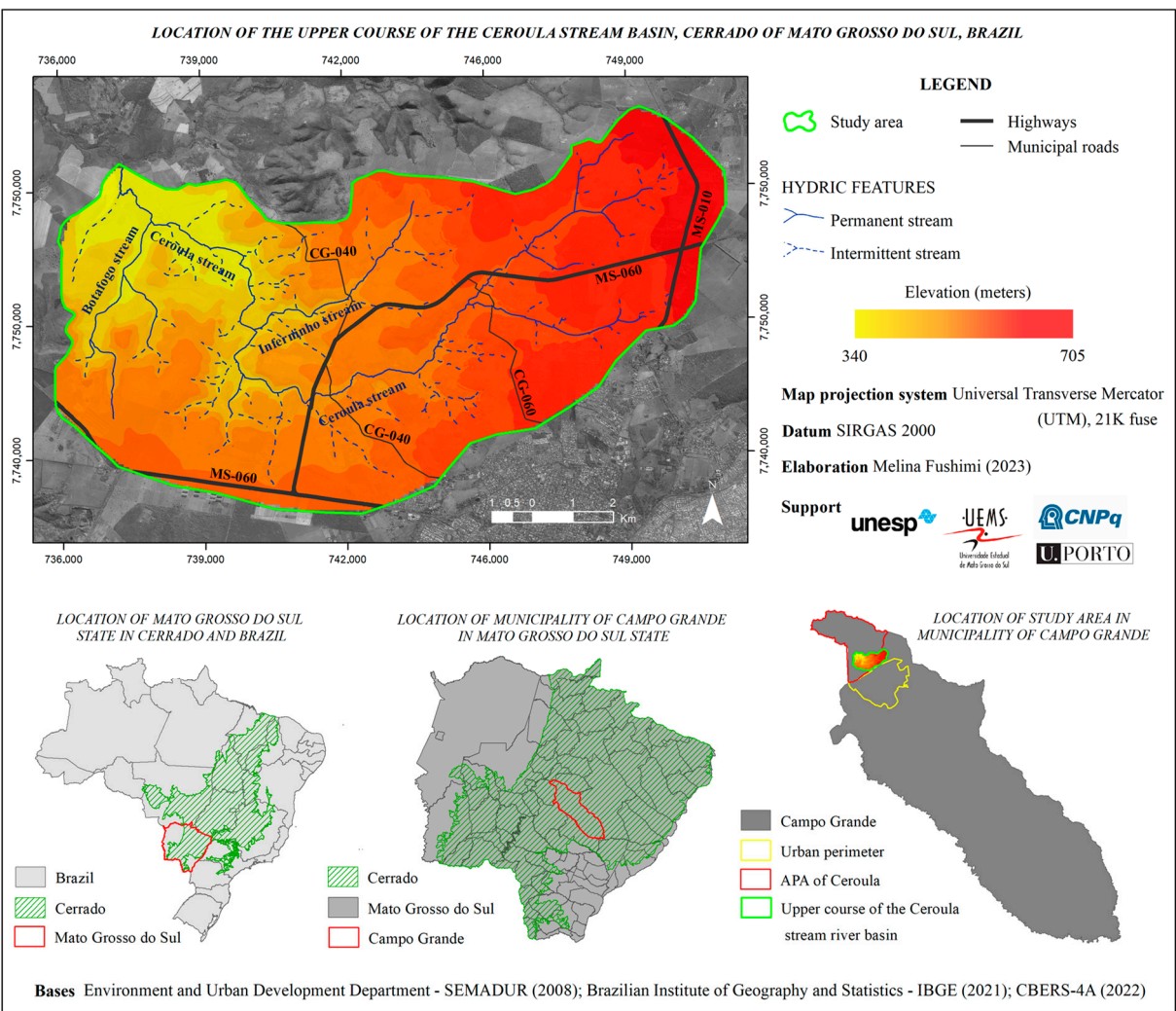

**Figure 1.** Location of the upper course of the Ceroula stream basin, Cerrado of Mato Grosso do Sul, Brazil.

## 3. Materials and Methods

For mapping land use and cover, Geographic Information System of ArcMap 10.5® software was used to manipulate satellite images. An image from Landsat 5 was used, from 13 August 1985, with a thematic mapper (TM) sensor and a spatial resolution of 30 m (Figure 2A). For the RGB color composition and enhancement of the spectral bands, the visible bands 5(R), 4(G), and 3(B) were chosen. In 2022, the image used is from CBERS-4A, dated July 19, 2022 using a wide panchromatic and multispectral camera (WPM). To increase the level of detail, the visible bands 3(R), 2(G), and 1(B) were merged with the panchromatic band 0, in which the spatial resolution increased from 8 m to 2 m (Figure 2B).

The choice of scenes for both the Landsat 5 and CBERS-4A images considered low cloud cover (up to 10%), and the images were made available free of charge through the National Institute for Space Research (INPE) catalogue [10]. Although the images have different spatial resolutions, both meet the objective of the manuscript, and the scale of the elaborated maps was 1:25,000.

The land use and cover classes (pasture, forest vegetation, exposed soil, temporary crop and others) were defined based on the adaptation of the nomenclature proposed by IBGE (2013) (Table 1), and the identification of land use and cover classes in the Landsat 5 (Figure 2A) and CBERS-4A (Figure 2B) images were made based on the supervised classification technique of ArcMap 10.5®.

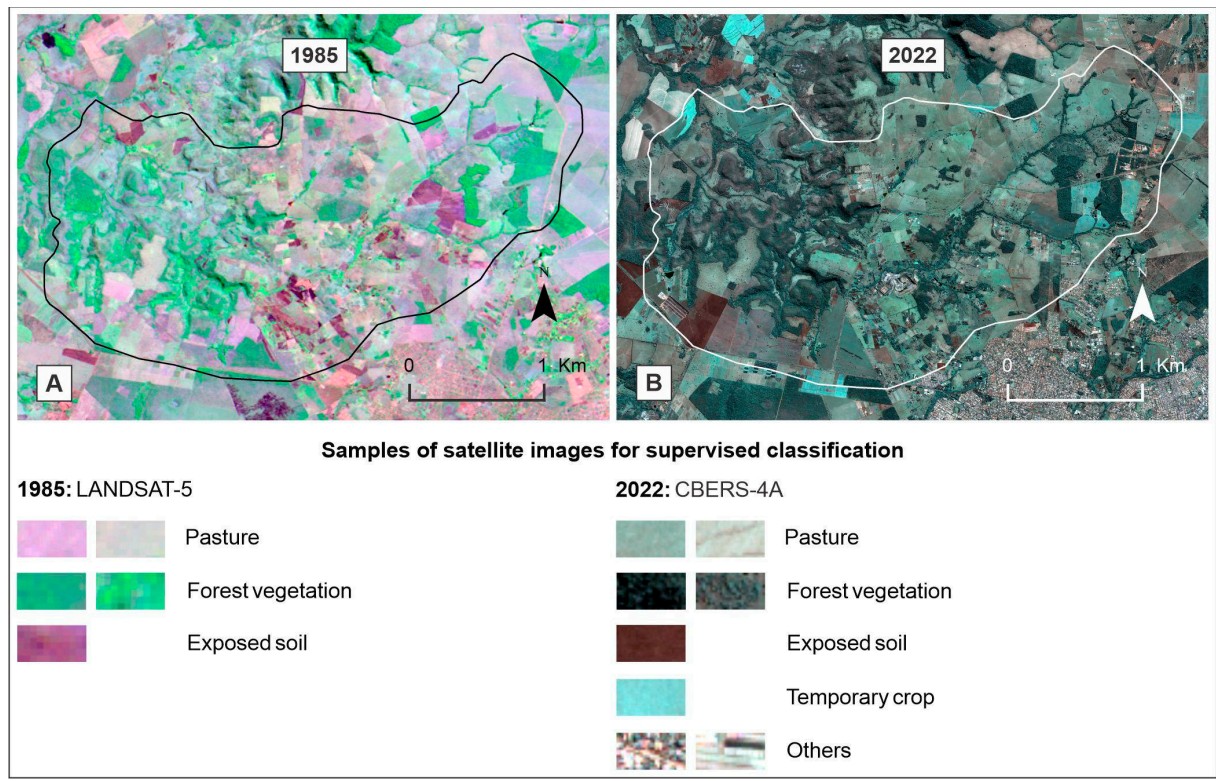

**Figure 2.** Samples of satellite images for supervised classification. (**A**): LANDSAT-5 image. (**B**): CBERS-4A image.

**Table 1.** Classes of land use and cover for the years 1985 and 2022.

| Classes of Land Use and Cover Proposed by IBGE (2013) | Classes of Land Use and Cover Adapted to the Study Area—1985 | Classes of Land Use and Cover Adapted to the Study Area—2022 |
|---|---|---|
| 1 Non-agricultural anthropogenic areas<br>1.1 Urbanized areas<br>1.2 Mining areas | - | Others |
| 2 Anthropogenic agricultural areas<br>2.1 Temporary crop<br>2.3 Pasture | Pasture | Temporary crop<br>Pasture |
| 3 Areas of natural vegetation<br>3.1 Forestry | Forest vegetation | Forest vegetation |
| 5 Other areas<br>5.1 Uncovered areas | Exposed soil | Exposed soil |

Supervised classification of maximum similarity was performed using the Maximum Likelihood Classification tool from ArcMap 10.5, and in the 2022 mapping, spatialized information was verified and validated in fieldwork through reambulation at 62 control points, using a global positioning system (GPS), camera, and drone.

In the 2022 land use and cover map, linear erosive features (rill, ravine, and gully), agricultural terraces, and retention basins were identified in the CBERS-4A image and confirmed in the fieldwork carried out in 2022 and 2023. The symbol for erosive features was adapted from Verstappen and Zuidam (1975) [11] and the representation of agricultural terraces was based on Verstappen and Zuidam (1975) [11].

The normalized difference vegetation index (NDVI) [12] is an important tool for identifying changes in land use and cover [13](Taiwo et al., 2023), especially in relation to summer and winter crops and the response of vegetation to seasonal variations in climate [14]. The NDVI was calculated for 2023 from images from the Sentinel-2 satellite,

with the red and near infrared bands for 15 dates. The images were obtained from the Copernicus hub and processed in Sentinel Application Platform (SNAP) software, version 9.0.0 [15]. This analysis is complementary to the classification of land use and cover and allows a detailed assessment of the health and density of vegetation cover throughout the year, enabling the identification of changes in land use and cover, such as livestock and soybean farming.

Precipitation data were obtained from the Royal Netherlands Meteorological Institute (KNMI) [16] and correspond to Climatic Research Unit Timeseries 4.0 (CRU TS4) [16] values, with a spatial resolution of $0.5 \times 0.5$ degrees and derived from an analysis of more than 4000 individual weather station records (Royal Netherlands Meteorological Institute, 2023) [16]. Although a homogenization process was carried out on many of the input records, it needs to be highlighted that the CRU TS4 series themselves are not strictly homogeneous. To ensure a more robust analysis, additional data were obtained from meteorological stations 83611 (Campo Grande, coordinates 737,576.99 m E and 7,737,377.87 m S) and 02154008 (Fazenda Ponte, coordinates 790,559.48 m E and 7,641,858.49 m S) from the Brazilian National Institute of Meteorology [17,18], located close to the study area. These data were used to establish correlations with the values presented by the CRU TS4 data series. It is important to mention that the INMET series have some limitations, such as reading errors, prolonged periods without data and low reliability of daily values. Therefore, these records were used as a confirmation of the values provided by CRU TS4.

## 4. Results and Discussion

The land use and cover between the years 1985 and 2022 of the upper course of the Ceroula stream basin, Cerrado of Mato Grosso do Sul, showed a decrease in pasture and exposed soil, an increase in forest vegetation, and the introduction of temporary crops (Figures 3 and 4, and Table 2).

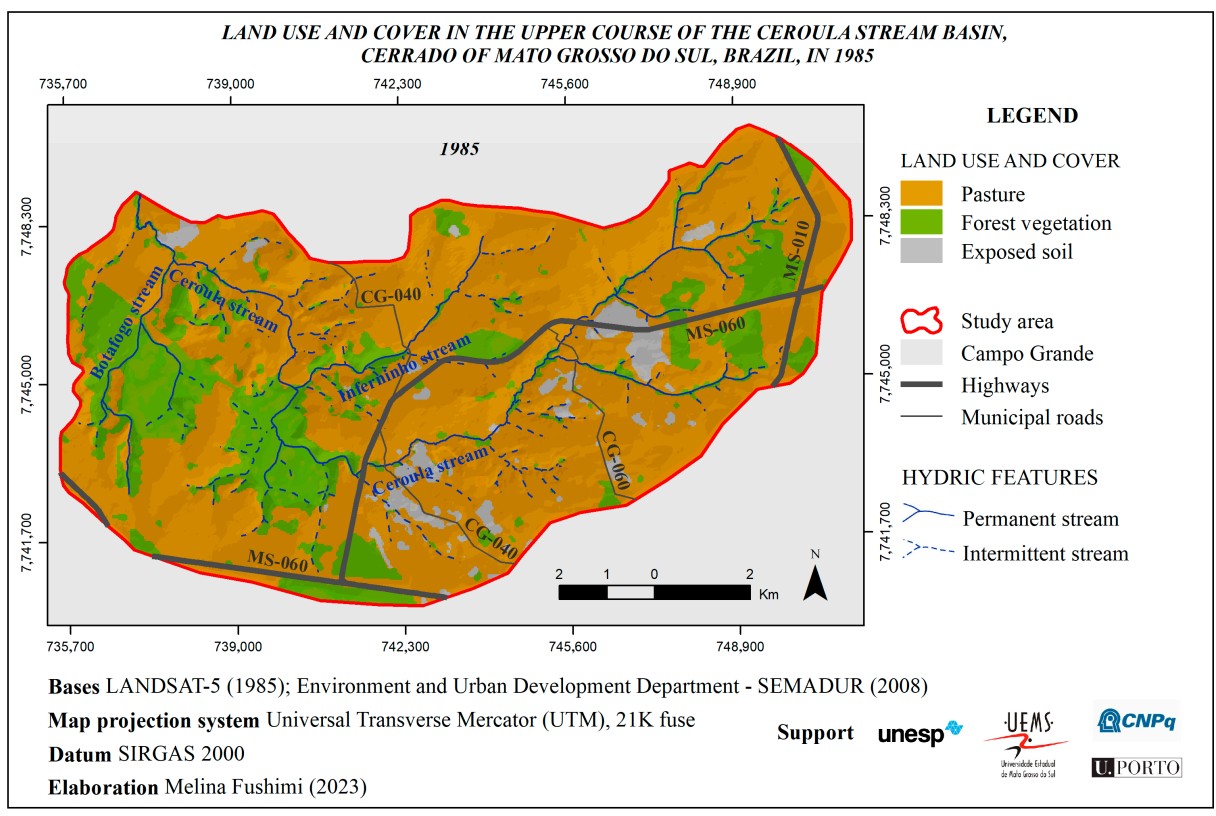

**Figure 3.** Land use and cover in the upper course of the Ceroula stream basin, Cerrado of Mato Grosso do Sul, Brazil, in 1985.

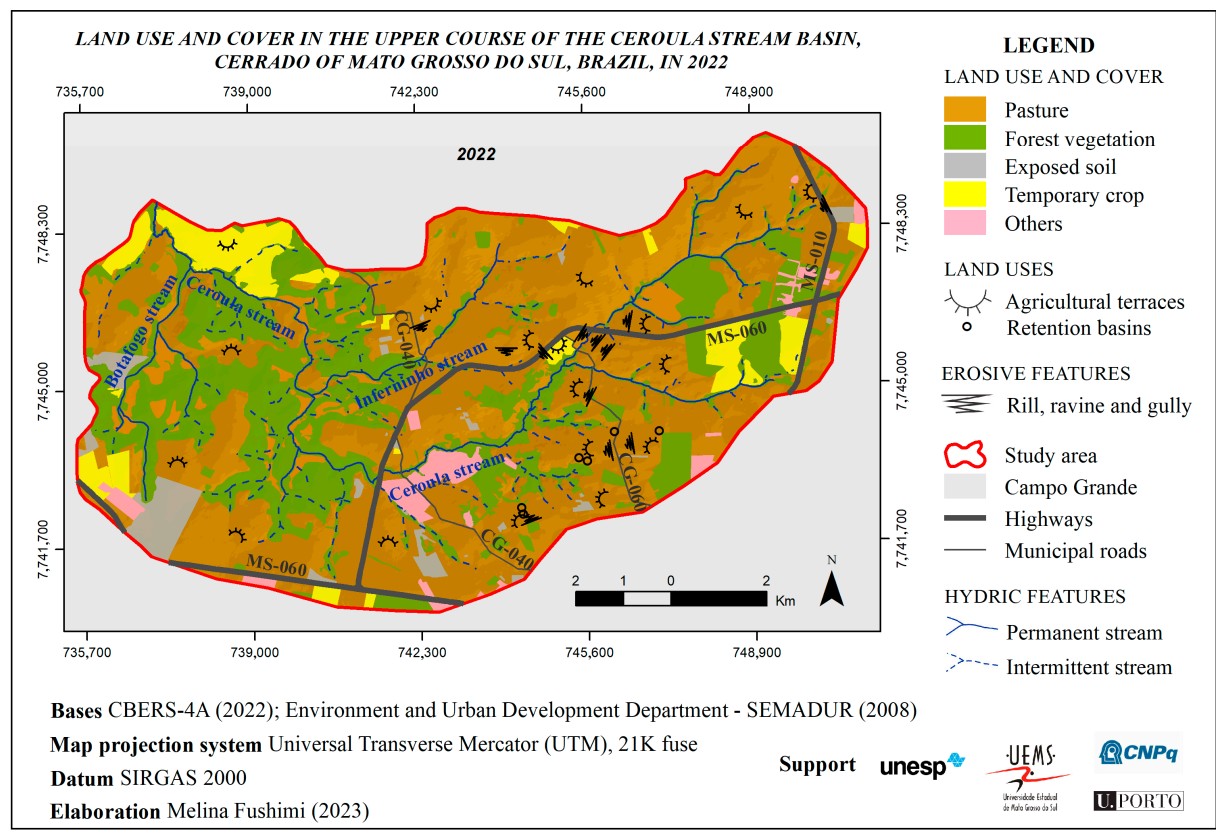

**Figure 4.** Land use and cover of the upper course of the Ceroula stream basin, Cerrado of Mato Grosso do Sul, Brazil, in 2022.

**Table 2.** Spatio-temporal evolution of land use and cover in the upper course of the Ceroula stream basin, Cerrado of Mato Grosso do Sul, Brazil, between the years 1985 and 2022.

| Land Use and Cover Classes | Area | | Area | | Area Variation (Δ) | |
|---|---|---|---|---|---|---|
| | 1985 | | 2022 | | 2022–1985 | |
| | km$^2$ | % | km$^2$ | % | km$^2$ | % |
| Pasture | 78.08 | 77.44 | 58.76 | 58.27 | −19.32 | −19.17 |
| Forest vegetation | 19.09 | 18.93 | 30.26 | 30.01 | 11.17 | 11.08 |
| Exposed soil | 3.66 | 3.63 | 2.96 | 2.93 | −0.70 | −0.70 |
| Temporary crops | | | 6.19 | 6.14 | 6.19 | 6.14 |
| Others | | | 2.66 | 2.65 | 2.66 | 2.65 |
| Total area | 100.83 | 100 | 100.83 | 100 | | |

Based on Figures 3 and 4, and Table 2, as well as information collected and reviewed in the fieldwork, there was an increase in forest vegetation from 19.09 km$^2$ in 1985 to 30.26 km$^2$ in 2022. This increase of 11.17 km$^2$ may be associated with the recovery of areas on the edge of abandoned pastures.

However, part of this vegetation cover is secondary, with the replacement of native species by invasive species, such as Leucaena (*Leucaena leucocephala*) (Figure 5A), which is located, preferably, close to urbanized areas and highways, where waste and pruning remains are disposed of. Fabricante (2014) [19] indicated that Leucaena has medium to high susceptibility to invasion in the Brazilian Cerrado, promoting the homogenization of the flora and affecting the resilience of invaded sites and productive arrangements (it reduces the quality of pastures and is a host to pests and crop diseases).

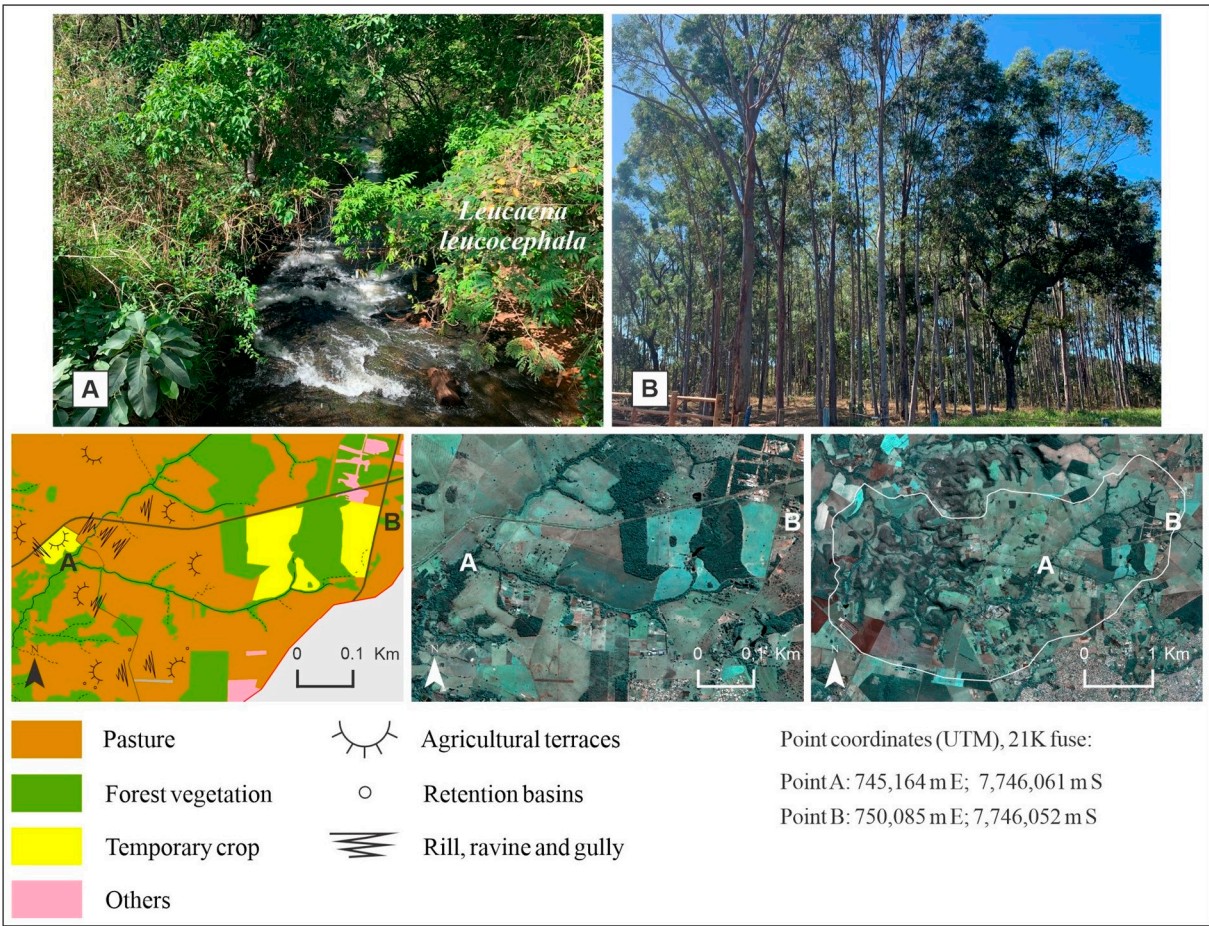

**Figure 5.** Land use and cover control points. Point (A): Leucaena (*Leucaena leucocephala*). Point (B): eucalyptus (*Eucalyptus* spp.).

Another aspect to be considered in this forest vegetation class is the forest species of eucalyptus (*Eucalyptus* spp.) for the cellulose market, being the most recent of the monocultures in the Brazilian Cerrado, including the Cerrado of Mato Grosso do Sul and the study area (Figure 5B). Behera and Sahani (2003) [20], when comparing the physicochemical characteristics of soils under native forest, regenerating forest, and eucalyptus plantation, indicated soils with eucalyptus cultivation as having low activity of microorganisms, due to harmful compounds released by eucalyptus litter and low carbon levels, and as a consequence, reduced structural stability and increased erosion potential.

The temporary crop in 2022 is produced for the international commodities market, with soybean (*Glycine max*) in the months of September to April or May (Figure 6A on 15 November 2022) and corn (*Zea mays*) in the off-season (Figure 6A on 6 June 2023).

Soybean monoculture is contextualized in the current process of expansion of the soybean agricultural frontier across the Brazilian Cerrado, incorporating the MATOPIBA region (states of Maranhão, Tocantins, Piauí, and Bahia) and advancing to other states, such as Mato Grosso do Sul [21,22]. Soybean production data by state in 25 harvests (1996–1997 to 2020–2021) from the Brazilian Agricultural Research Corporation [23] indicated that Mato Grosso presented the highest growth in annual production (higher than 1257 thousand tons per year), followed by Rio Grande do Sul, Paraná, Goiás, and Mato Grosso do Sul, with annual rates of 611.9, 590.3, 412.8, and 362 thousand tons, respectively.

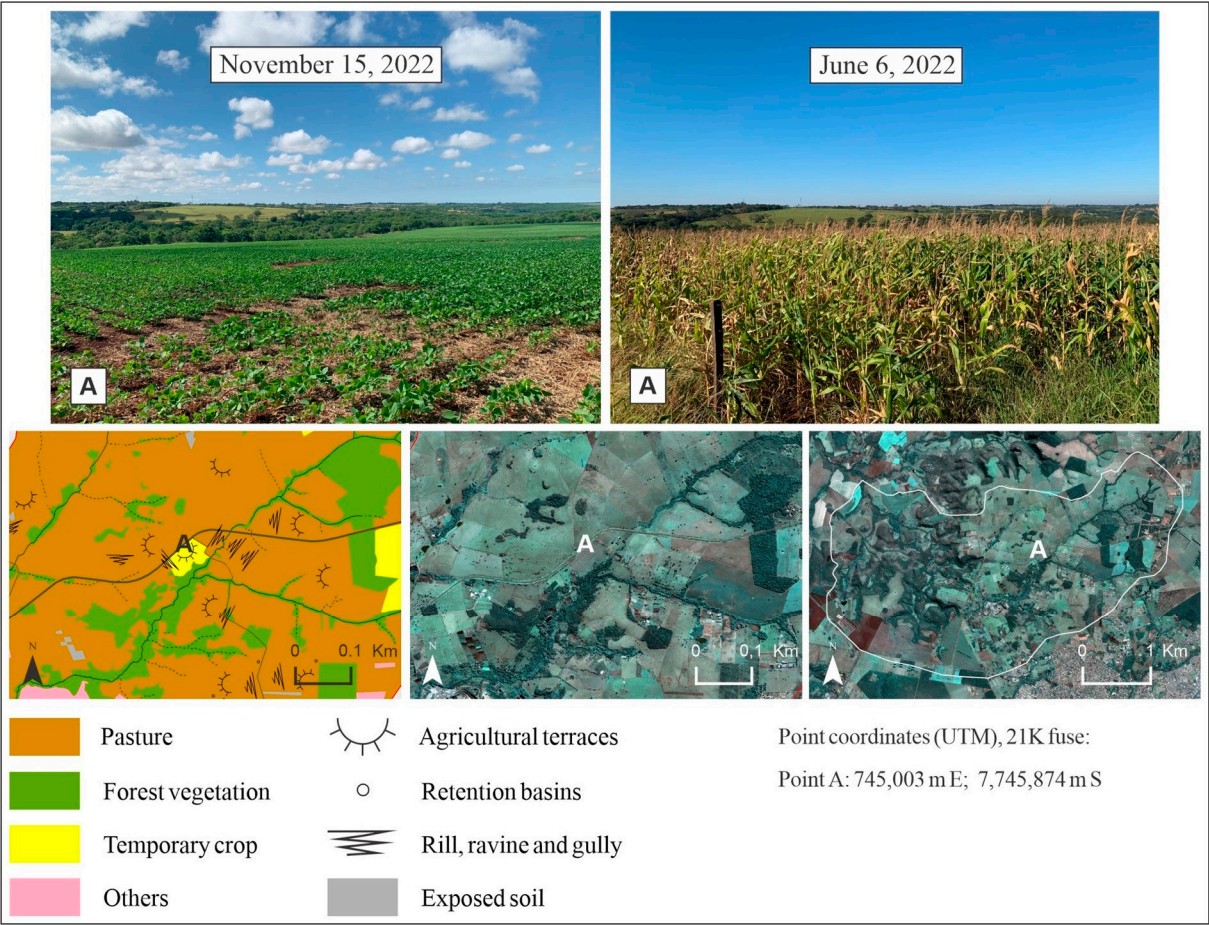

**Figure 6.** Land use and cover control points. Point (A) on November 15, 2022: soybean (*Glycine max*). Point A on 6 June 2022: corn (*Zea mays*) in the off-season.

In Mato Grosso do Sul state, Campo Grande stands out as one of the municipalities with values above the state average of productivity throughout the 2022–2023 soybean harvest, with productivity of 68.37 sc/ha, in an area of 112,931.20 ha, and production of 463,242.52 tons [24]. Although the process of incorporation of the soybean complex in Mato Grosso do Sul state began in the 1970s and 1980s, its introduction in the municipality of Campo Grande, where the upper course of the Ceroula stream basin is located, occurred later, from 2014 [25,26]. Thus, the temporary crop class consisting of soybeans and corn is not spatialized on the land use and cover map for 1985, only for 2022, with an extension of 6.19 km$^2$.

Hernani et al. (1997) [27] demonstrated that the soil loss in soybean and corn planting sectors occurs mainly in the period between soil preparation and grain sowing, constituting a temporary condition of exposed soil. Therefore, the detachment and transport of sediments that characterize the linear erosive dynamics develop under these circumstances of exposed soil that extended 2.96 km$^2$ in the study area on 19 July 2022 (date of the CBERS-4A image).

In addition to physical soil degradation through erosion, in the production of soybeans and corn, corrective agents, pesticides, fertilizers, agricultural machines, and equipment are used to maintain/increase production, which can lead to deep chemical contamination and compaction of the ground [28,29]. Henrique et al. (2021) [30], when comparing two units with Cerrado soils, one with soybean cultivation and the other as an environmental reserve, found that the formation of compacted layers was more intense in mechanized soybean planting, especially in areas with longer agricultural use, greater than 10 years. In Mato Grosso do Sul state, Bombardi (2019) [31] estimates that the annual average pesticide

use is 51,534 tons, and that in the municipality of Campo Grande, 16.85% to 25.33% of all establishments use pesticides.

Table 3 presents the products of temporary crops and their planted area in the municipality of Campo Grande for 2021, with an emphasis on soybeans (940 km$^2$ of planted area), followed by corn in the off-season (600 km$^2$ of planted area), destined for the agribusiness [32].

**Table 3.** Temporary crops and planted area in the municipality of Campo Grande, Cerrado of Mato Grosso do Sul, Brazil, in 2021.

| Temporary Crop | Planted Area (km$^2$) |
|---|---|
| Pineapple | 0.3 |
| Herbaceous cotton (seed) | 6.13 |
| Sweet potato | 0.02 |
| Sugarcane | 30 |
| Beans (grain) | 1 |
| Cassava | 2.9 |
| Watermelon | 0.27 |
| Corn (grain) | 600 |
| Soybeans (grain) | 940 |
| Sorghum (grain) | 30 |
| Tomato | 0.02 |

Source: IBGE—Municipal Agricultural Production (2022).

The other class included in the mapping of 2022, with an area of 2.66 km$^2$, covers different types of land use and cover after 1985, among which are basalt from the Serra Geral Formation [8] and sand extraction from the Caiuá Formation [8], both activities being aimed at the internal trade for civil construction in the Campo Grande region. The basalt is exposed and disaggregated using machinery (Figure 7A), while the sand is extracted using equipment such as a backhoe and dredger (Figure 7B). In both situations, the vegetation is completely removed and there is significant detachment and transport of material.

Analyzing the environmental implications of a mining area in southwest Spain, Duque et al. (2015) [33] indicate that the impact of rain on weakly consolidated mining deposits, without vegetation cover and exposed to erosive agents, promotes runoff and various erosive features (earth pillar, rill, ravine, and gully). The land use for mining activities, without vegetation and associated with rainfall, triggers accelerated pluvial erosion, as observed in the upper course of the Ceroula stream basin.

The other category also includes urbanized areas, expanding over pasture and forest vegetation, located in an urban expansion zone that allows the installation of new industrial and mining activities and aims to integrate urban and rural activities [25]. However, this has consequences in reducing infiltration and increasing runoff and erosive potential. In addition, erosion associated with the urbanization of Campo Grande is a phenomenon that has been occurring in other river basins in the municipality, such as the Prosa [34,35] and Guariroba streams [36].

Finally, the other class includes rural communities that raise animals and produce various agricultural foods, for example, fruits, vegetables, cassava, and free-range chickens. These were not included in temporary crops and pasture, as they are local subsistence practices on small properties, aimed at the region's internal market, with less environmental degradation compared to crops for the international commodities market, which are produced on large estates without adequate environmental conservation measures.

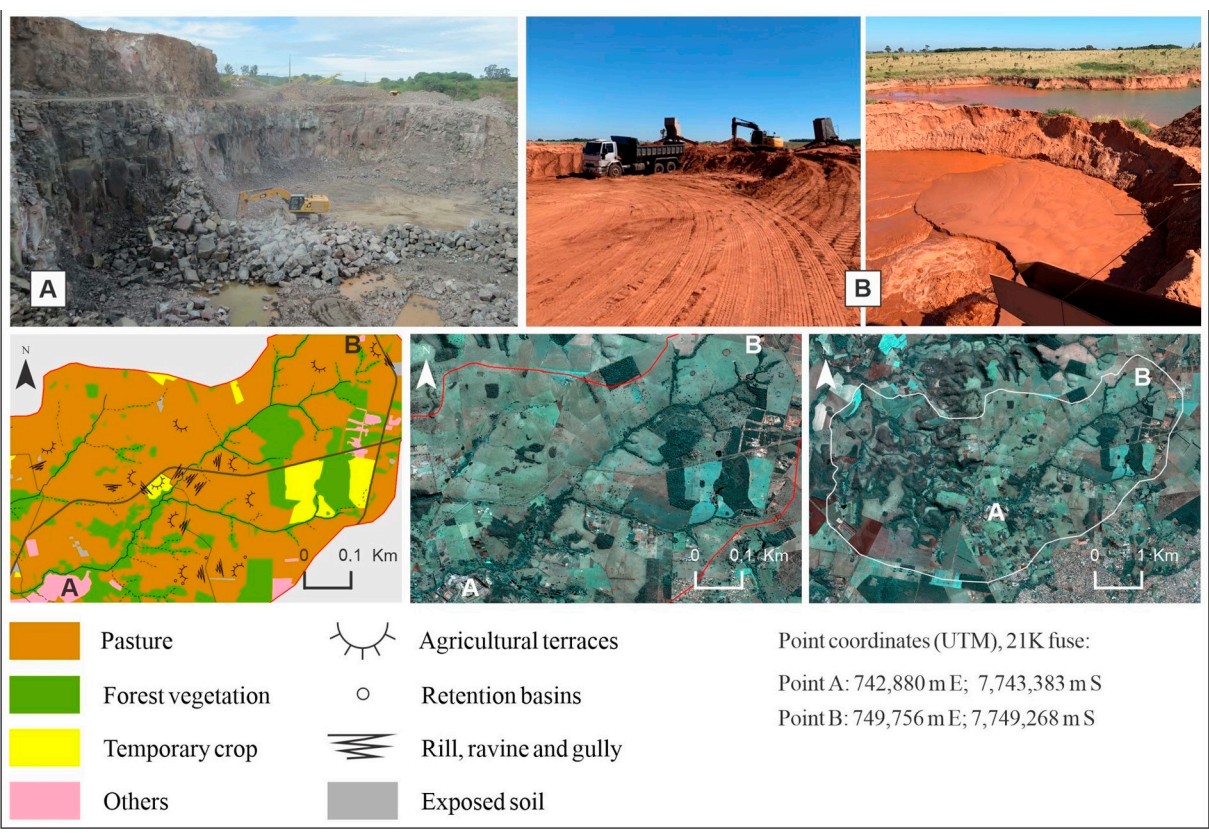

**Figure 7.** Land use and cover control points. Point (A): basalt extraction. Point (B): sand extraction.

Despite the particularities of land use and cover existing between mining, urbanized areas, and small properties, and their different influences on the development of linear erosion, these were classified into a single category due to their occurrence at specific points. In addition, given the scale of the analysis (1:25,000), they are not cartographically representative if classified separately.

The oil crisis in the 1970s impacted the regional landscape of the Cerrado of Mato Grosso do Sul, replacing the wheat/pasture with soybean/pasture, in order to reduce the consequences of the international instability scenario [26]. In the upper course of the Ceroula stream basin, pasture is the most significant land use and cover, with 78.08 km$^2$ in 1985 and 58.76 km$^2$ in 2022. Comparing these years, the reduction of 19.32 km$^2$ is associated, especially, with the gradual introduction of soybean from 2014 and with the urban expansion in the east–south sector of the study area.

Used as food for cattle in extensive and intensive systems and for dairy farming, the vegetation is Brachiaria grass, and in many of the pastures, there are artificial terraces (Figure 8A). Agricultural terracing is a mechanical erosion control practice and aims to retain and infiltrate rainfall on level terraces or slowly drain it to adjacent areas [37]. The structure is composed of a dike and a channel that is transverse to the slope [38].

Linear erosive processes were observed in pasture sectors (Figure 8B) in which the structures were not maintained, with the ridges being broken by cattle. The rupture of terraces can also harm other structures located in the base, causing great damage to the cultivated area [39].

In the pastures close to the CG-060 Municipal Road, in addition to terracing, there are retention basins that have the function of retaining rainfall and reducing erosion close to the roads. However, in some sectors, the maintenance of conservation structures does not occur, and consequently linear erosive processes (rill, ravine and gully) are installed. According to Table 4, although agricultural terraces and retention basins are preferably located in pasture areas, the 12 linear erosive features identified in the study area occur in

these pastures. Therefore, the adoption of complementary conservation measures, such as vegetative and edaphic actions, are also relevant.

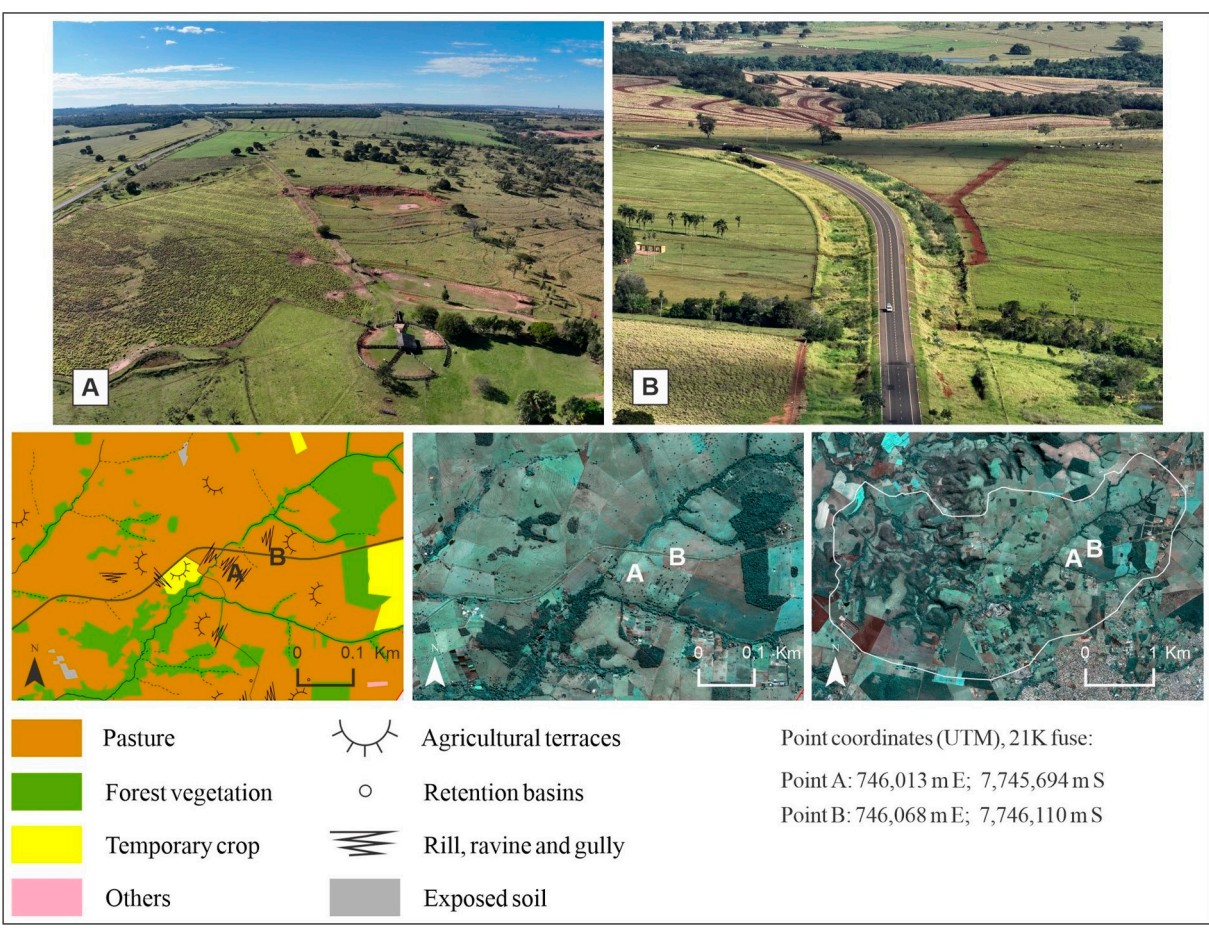

**Figure 8.** Land use and cover control points. Point (A): grass for pasture and erosion. Point (B): artificial terraces and erosion.

**Table 4.** Agricultural terraces, retention basins, and erosive features by land use and cover classes.

| Land Use and Cover Classes | Agricultural Terraces | Retention Basins | Erosive Features |
| --- | --- | --- | --- |
| Pasture | 16 | 6 | 12 |
| Forest vegetation | 0 | 0 | 0 |
| Exposed soil | 0 | 0 | 0 |
| Temporary crop | 2 | 0 | 0 |
| Others | 0 | 0 | 0 |
| Total | 18 | 6 | 12 |

In the Brazilian Cerrado, Peron and Evangelista (2004) [40] estimated that 80% of the 50 to 60 million ha of pasture are in some stage of degradation and Greschuk (2022) [41] estimated that degradation levels vary from moderate to severe. Given the climate and soil conditions that define less vigorous native vegetation, Dias-Filho (2014) [42] indicated that the most frequent type of pasture degradation in the Cerrado is biological, in which the soil significantly loses its capacity to sustain plant production, leading to the replacement of pasture by plants that are not demanding on soil fertility or to the appearance of areas devoid of vegetation (exposed soil). In the upper course of the Ceroula stream basin, although it was not quantified, in the fieldwork it was observed that pastures with the

absence of agricultural terraces or without maintenance of structures and with linear erosive features have low vigor, with invasive plants and areas of exposed soil.

The erosive process intensifies during the rainy season, between the months of October and March, when a greater concentration of rainfall occurs in the study region—approximately 73% of the annual total (Figure 9). During this period, there is a greater frequency of days with precipitation and the occurrence of intense rain showers and thunderstorms is more common. Rainfall is generally concentrated in short periods, increasing the kinetic energy of the water and its ability to detach and transport sediments [43–46].

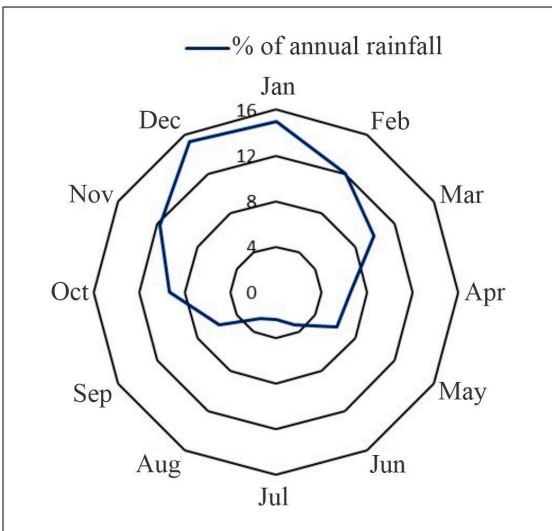

**Figure 9.** Monthly rainfall concentration in the upper course of the Ceroula stream basin, Cerrado of Mato Grosso do Sul, Brazil. Source: INMET (2023); Royal Netherlands Meteorological Institute (2023).

In 2023, the highest volume of rainfall in Campo Grande occurred in the summer (January to March) and autumn (March to June) months, reflecting on the regrowth and vigor of the vegetation, as demonstrated by the normalized difference vegetation index on 27 February, 29 March, 3 April, 13 May, and 7 June 2023 (Figure 10). However, areas with low vegetation vigor are preferably located in temporary crop sectors. On 27 February, the soybeans were close to harvest and stubble areas were being prepared for planting off-season corn. In the image from 29 March, some soybean stubble areas still have exposed soil, and in others, corn is in full development. In the fieldwork, it was found that in some areas cultivated with soybeans in the summer, winter pastures were planted, such as corn. At the peak of plant growth, which occurred in April, the land use and cover with grass for pasture indicated high vegetation vigor.

During the winter months (June to September), vegetation vigor was expected to decrease (Figure 10, on 22 July; 1, 6, and 31 August; and 10, 15, and 20 September 2023), due to the reduction in precipitation and soil moisture, in addition to the deciduousness of some Cerrado phytophysiognomies. In this dry season, it is possible to observe extensive areas with exposed soil that correspond to pasture areas being converted into soybean crops (temporary crop), pastures under renovation, and areas used for vegetable production. When the rains return, the deciduous plants in the forest phytophysiognomy resprout and produce green leaves again. In pasture and crop areas, NDVI values also increase in response to the meteorological factor precipitation. On all dates in 2023, it is possible to observe the water mirror in mining pits (basalt and sand extraction), reservoirs formed by the construction of earthen dams, and pisciculture tanks in humid riverine areas.

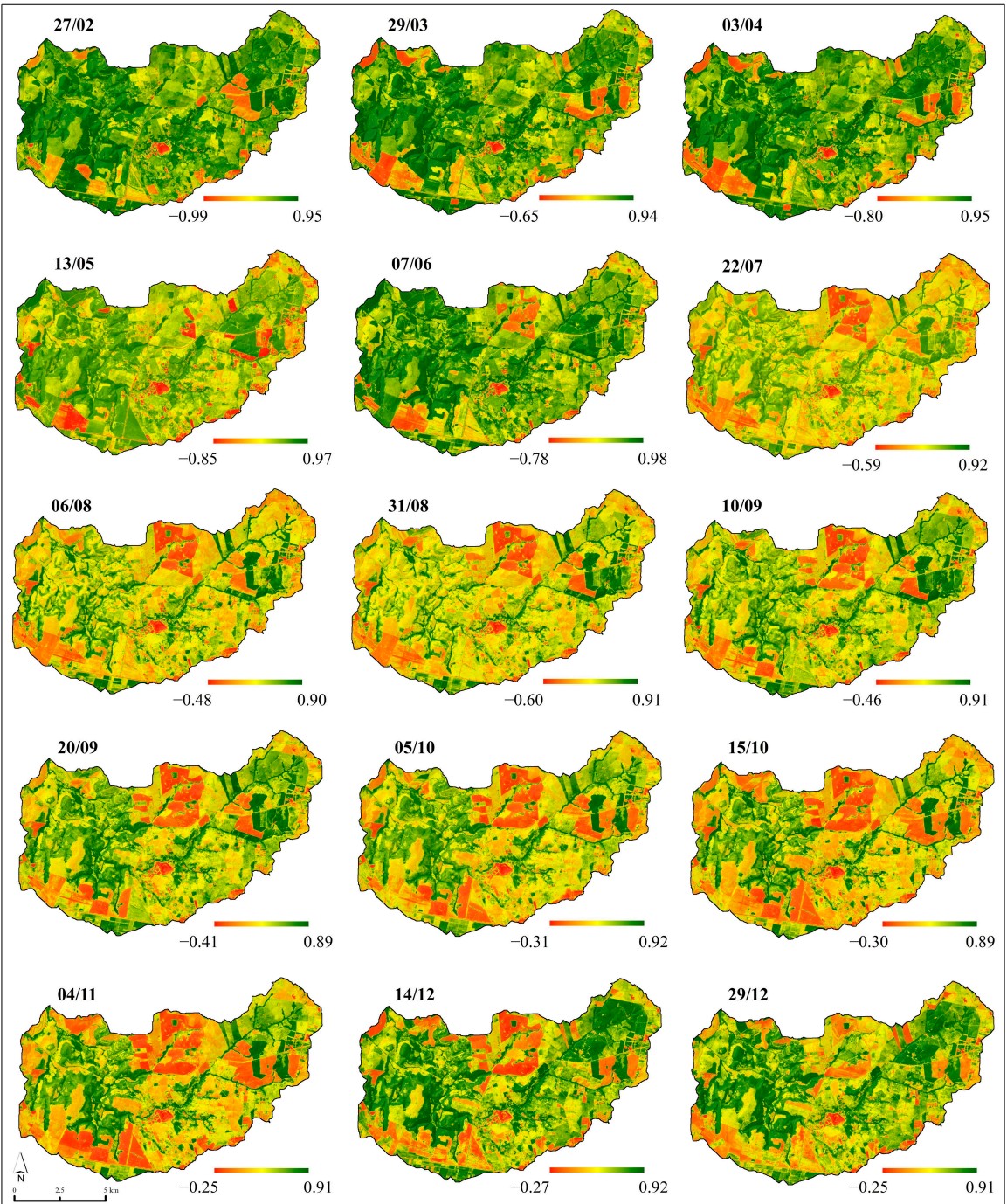

**Figure 10.** Normalized difference vegetation index for 2023. Source: ESA, Copernicus (2023).

The histograms in Figure 11 quantitatively express changes in vegetation phenology and in land use and cover throughout 2023.

From the mapping and fieldwork, changes in land use and cover were verified in the upper course of the Ceroula stream basin, with spatio-temporal changes in the Cerrado of Mato Grosso do Sul. Pasture and the introduction of temporary crops stand out, with the production of grains (soybean) gradually advancing and destroying the original forest vegetation of the Cerrado. Thus, nature is appropriated as a resource, strongly influenced by the international commodities market, without adequate management and with the absence of conservation practices, providing accelerated linear erosive processes (rill, ravine, and gully) in the landscape.

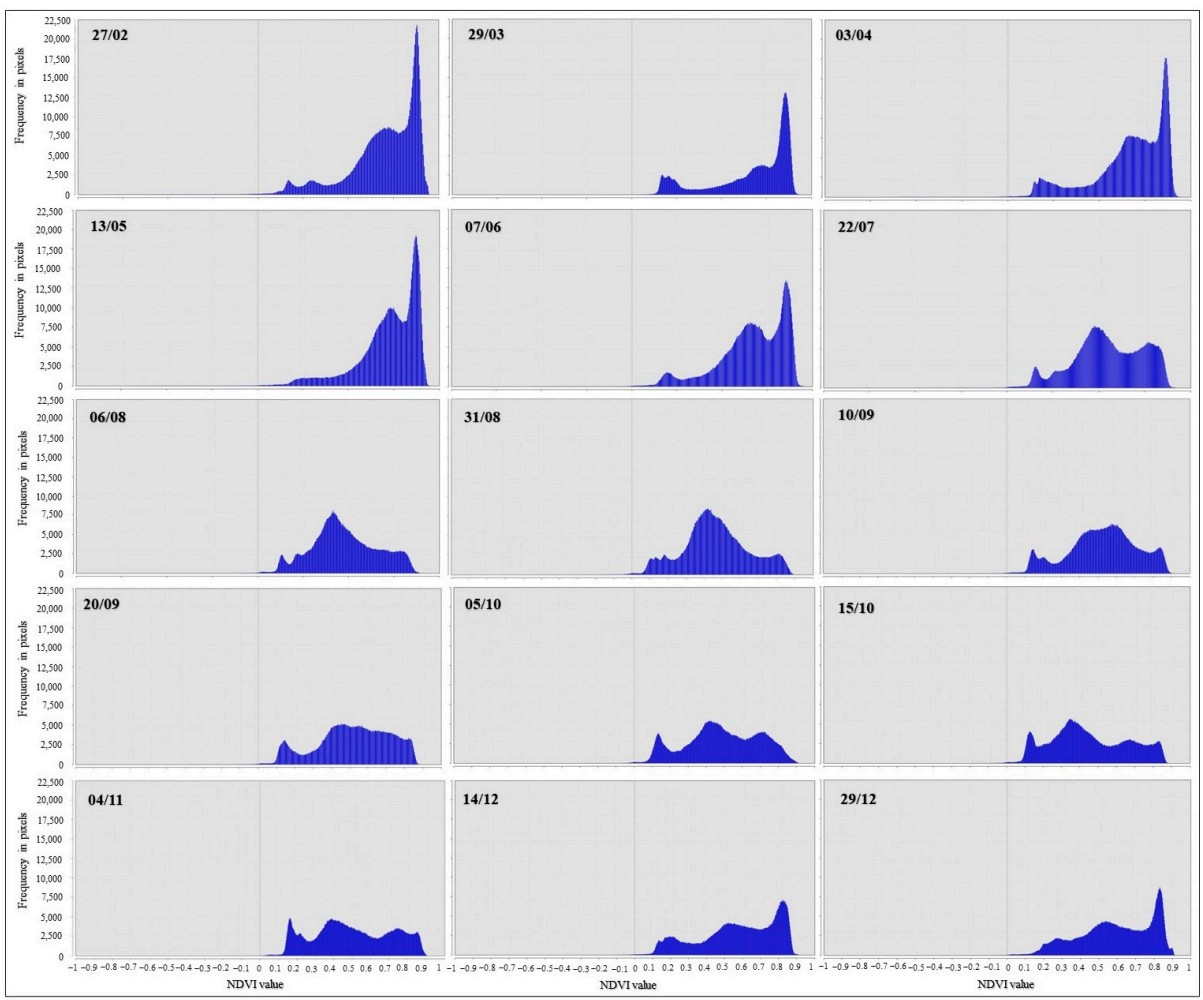

**Figure 11.** Histogram of normalized difference vegetation index for 2023. Source: ESA, Copernicus (2023).

## 5. Conclusions

The key results from this research are as follows.

(1) Fieldwork is important to analyze and record the dynamics of land use and coverage on a local scale and to update mappings.

(2) The upper course of the Ceroula stream basin underwent intense environmental transformations between 1985 and 2022. In 1985, forest vegetation was replaced by pasture, and in 2022, in addition to pasture, there was the introduction of soybean monoculture.

(3) Although urban areas are not very representative, the research area is a sector of urban expansion in the city of Campo Grande, and thus land use and cover may undergo significant changes in the coming years.

(4) The study area, despite being part of a sustainable use conservation unit, is under strong interest from the international commodities market. Thus, sustainable environmental management is necessary, together with public policies, in order to avoid or minimize the degradation processes and environmental impacts.

These results may help provide important insights into the dynamics of land use and cover, the consequences of the lack of conservation practices, and the environmental impacts in the Cerrado of Mato Grosso do Sul, contributing to better understanding of the environmental challenges faced in the region and encouraging subsidies for the development of sustainable management strategies.

**Author Contributions:** Conceptualization, M.F.; methodology, M.F.; software, M.F.; validation, M.F.; formal analysis, M.F.; research, M.F., G.N.d.L. and V.C.; writing—original draft, M.F., G.N.d.L. and V.C.; writing—revision and editing, G.N.d.L. and V.C.; visualization, G.N.d.L. and V.C.; fund acquisition, M.F. All authors have read and agreed to the published version of the manuscript.

**Funding:** This work received support from the National Council for Scientific and Technological Development (CNPq) through the Research Productivity Grant (PQ)—Level 2, Process 308723/2021-0, as well as from the Centre of Studies in Geography and Spatial Planning (CEGOT), funded by national funds through the Foundation for Science and Technology (FCT) under the reference UIDB/04084/2020.

**Institutional Review Board Statement:** Not applicable.

**Informed Consent Statement:** Not applicable.

**Data Availability Statement:** Data are contained within the article.

**Conflicts of Interest:** The authors declare no conflicts of interest.

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
