# Peer review of "Changes in Land Use and Cover and Their Environmental Impacts in the Cerrado of Mato Grosso Do Sul, Brazil"

_sustainability, doi:10.3390/su16104266_

Round 1

Reviewer 1 Report

Comments and Suggestions for Authors

This paper analyzed the changes in land use and cover and their environmental impacts in the Cerrado of Mato Grosso Do Sul, Brazil. The methods are dependable. The conclusions are convinced. It contributed to better understanding of the environmental challenges faced in the region. However, the part of results and discussions are too long and I have no clue to which questions is important for this paper. I suggest that the discussion was separated from this part and focused on the main scientific questions you are ready for answering. In addition, How did you calculate the Normalized difference vegetation index, what does it means?

Comments on the Quality of English Language

The language should be improved by special institution.

Author Response

We greatly appreciate your attentive and careful review of our manuscript. Below we specify the corrections made based on your review:

- The part of results and discussions are too long and I have no clue to which questions is important for this paper. I suggest that the discussion was separated from this part and focused on the main scientific questions you are ready for answering.

The consideration is relevant. However, the results are reinforced by other scientific research carried out in the area of the study or in the Cerrado region of Mato Grosso do Sul. Therefore, we chose to report the results and discussion together to facilitate understanding of the study.

- How did you calculate the Normalized difference vegetation index, what does it means?

The NDVI is an important tool for identifying changes in land use and cover, especially in relation to summer and winter crops and the response of vegetation to seasonal climate variations. The NDVI was calculated for the year 2023 from images from the Sentinel-2 satellite, with the Red and Near Infrared bands for 15 dates. The images were obtained at the Copernicus hub and processed in Sentinel Application Platform (SNAP) software, version 9.0.0. This analysis is complementary to the classification of land use and cover and allows a detailed assessment of the health and density of vegetation cover throughout the year, enabling the identification of changes in land use and cover, such as livestock-soybean farming. These explanations have been inserted in the revised text.

- The language should be improved by special institution.

The text was reviewed by a specialized English translator; please see the attached translation certificate.

All changes made to the text, based on the suggested revisions, are highlighted in blue.

Reviewer 2 Report

Comments and Suggestions for Authors

* The abstract section of the study needs to be rearranged. To ensure the integrity of the meaning, the paper should be read many times by different people and optimized. There is intense confusion in the meaning in ". The results showed spatio-temporal alterations between the years analyzed, with the introduction of soybean monoculture with corn in the off-season, the practice of which is contextualized in the current process of expansion of the soybean  agricultural frontier across the Brazilian Cerrado. There was also maintenance of pastures in the  grass sectors consisting of brachiaria and, despite the implementation of agricultural terracing and catchment basins, linear erosive features of the types furrows, ravines, and gullies were identified in this class of use and cover. When considering the importance of the Brazilian Cerrado and the study area that integrates the Cerrado of Mato Grosso do Sul and a Sustainable Use Conservation Unit, the research demonstrated that the upper course of the Ceroula stream river basin has interests economic and is under strong influence from the international commodities market."
* There is intense ambiguity in the article until the end of the introduction section. and this confusion of meaning continues throughout the rest of the article.

*Add label to Figure 1.
* considering all kinds of readers, an explanation of each method should be provided. For example, it should be stated for what purpose applications such as NDVI are used.

*167-170, 186-189 not clear. 
*228-231 not clear.

* figure 5,6 add label and explanation.
*Table 3: either use ha or km^2 to make it consistent throughout the paper.
*The conclusion section needs to be rearranged.

Comments on the Quality of English Language

*it's important to pay attention to English proficiency to ensure the paper is clear and effective. This means simplifying complex sentences, avoiding jargon, and providing clear explanations for a broader audience. Additionally, maintaining proper grammar, punctuation, and sentence structure will make the reading experience smoother

Author Response

We greatly appreciate your attentive and careful review of our manuscript. Below we specify the corrections made based on your review:

- The abstract section of the study needs to be rearranged.  

The section has been reorganized to provide clear, summarized information about the study.

- To ensure the integrity of the meaning, the paper should be read many times by different people and optimized.

- There is intense ambiguity in the article until the end of the introduction section. and this confusion of meaning continues throughout the rest of the article.

The introduction section has been reorganized to provide an unambiguous explanation. The remainder of the article has also been revised to correct confusing terms.

- Add label to Figure 1.

The label has been inserted in Figure 1.

- Considering all kinds of readers, an explanation of each method should be provided. For example, it should be stated for what purpose applications such as NDVI are used.

The NDVI is an important tool for identifying changes in land use and cover, especially in relation to summer and winter crops and the response of vegetation to seasonal climate variations. The NDVI was calculated for the year 2023 from images from the Sentinel-2 satellite, with the Red and Near Infrared bands for 15 dates. The images were obtained at the Copernicus hub and processed in Sentinel Application Platform (SNAP) software, version 9.0.0. This analysis is complementary to the classification of land use and cover and allows a detailed assessment of the health and density of vegetation cover throughout the year, enabling the identification of changes in land use and cover, such as livestock-soybean farming. These explanations have been inserted in the revised text.

- 167-170, 186-189 not clear.

228-231 not clear.

Lines 167-170, 186-189 and 228-231 have been revised to provide clear explanations.

- Figure 5,6 add label and explanation.

The label and explanation have been inserted in Figures 5 and 6.

- Table 3: either use ha or km2 to make it consistent throughout the paper.

km2 is used in Table 3. Data in ha have been excluded.

- The conclusion section needs to be rearranged.

The conclusion section has been reorganized into topics in order to provide clear conclusions and simplify complex sentences.

- It's important to pay attention to English proficiency to ensure the paper is clear and effective. This means simplifying complex sentences, avoiding jargon, and providing clear explanations for a broader audience.

The text has been revised to simplify complex sentences and provide clear explanations, avoiding regional jargon.

Additionally, maintaining proper grammar, punctuation, and sentence structure will make the reading experience smoother.

The text was reviewed by a specialized English translator; please see the attached translation certificate.

All changes made to the text, based on the suggested revisions, are highlighted in blue.

Reviewer 3 Report

Comments and Suggestions for Authors

This article aims to analyze changes in land use and cover in the upper reaches of the Ceroula stream watershed, located in the Cerrado of Mato Grosso do Sul, Brazil. The Geographic Information System was used to process satellite images and map land use and cover from the years 1985 and 2022. Based on the fieldwork, the spatialized information on the 2022 map was verified and validated through reambulation at control points.

This paper has strong innovation, rich research content, and certain significance. Suggest making the following improvements.

Firstly, how to demonstrate sustainability and better align with the theme of the journal?

Secondly, the research review did not reflect land use and cover; Soybean; Pasture; Erosion.

Thirdly, detailed research methods need to be supplemented, and the current version is more like a research report.

Fourthly, it is necessary to supplement research and discussion to reflect the contribution of this article.

Comments on the Quality of English Language

Extensive editing of English language required

Author Response

We greatly appreciate your attentive and careful review of our manuscript. Below we specify the corrections made based on your review:

- How to demonstrate sustainability and better align with the theme of the journal?

The research aims to provide important insights into the dynamics of land use and cover, the consequences of the lack of conservation practices and the environmental impacts in the Cerrado of Mato Grosso do Sul, contributing to the environmental diagnosis and better understanding of the environmental challenges faced in the region, demonstrating the importance of developing sustainable management strategies. These explanations have been inserted in the revised text.

- The research review did not reflect land use and cover; Soybean; Pasture; Erosion.

The keywords “Soybean”, “Pasture” and “Erosion” have been replaced by “environmental impacts” which are environmental consequences of land use and cover without sustainable management.

- Detailed research methods need to be supplemented, and the current version is more like a research report.

The methods have been supplemented, especially the NDVI. The NDVI is an important tool for identifying changes in land use and cover, especially in relation to summer and winter crops and the response of vegetation to seasonal climate variations. The NDVI was calculated for the year 2023 from images from the Sentinel-2 satellite, with the Red and Near Infrared bands for 15 dates. The images were obtained at the Copernicus hub and processed in the Sentinel Application Platform (SNAP) software, version 9.0.0. This analysis is complementary to the classification of land use and cover and allows a detailed assessment of the health and density of vegetation cover throughout the year, enabling the identification of changes in land use and cover, such as livestock-soybean farming. These explanations have been inserted in the revised text.

- Extensive editing of English language required.

The text was reviewed by a specialized English translator; please see the attached translation certificate.

All changes made to the text, based on the suggested revisions, are highlighted in blue.

Round 2

Reviewer 1 Report

Comments and Suggestions for Authors

The revised version has been improved a lot. I have no other suggestions.

Reviewer 2 Report

Comments and Suggestions for Authors

Thanks for the revision.

Reviewer 3 Report

Comments and Suggestions for Authors

Well done